# Single Neural Adaptive PID Control for Small UAV Micro-Turbojet Engine

**DOI:** 10.3390/s20020345

**Published:** 2020-01-08

**Authors:** Wei Tang, Lijian Wang, Jiawei Gu, Yunfeng Gu

**Affiliations:** School of Automation, Northwestern Polytechnical University, Xi’an 710129, China; wanglijian@mail.nwpu.edu.cn (L.W.); 026gjw@mail.nwpu.edu.cn (J.G.); guyunfeng@mail.nwpu.edu.cn (Y.G.)

**Keywords:** micro-turbojet engine, neural networks, adaptive control, PID

## Abstract

The micro-turbojet engine (MTE) is especially suitable for unmanned aerial vehicles (UAVs). Because the rotor speed is proportional to the thrust force, the accurate speed tracking control is indispensable for MTE. Thanks to its simplicity, the proportional–integral–derivative (PID) controller is commonly used for rotor speed regulation. However, the PID controller cannot guarantee superior performance over the entire operation range due to the time-variance and strong nonlinearity of MTE. The gain scheduling approach using a family of linear controllers is recognized as an efficient alternative, but such a solution heavily relies on the model sets and pre-knowledge. To tackle such challenges, a single neural adaptive PID (SNA-PID) controller is proposed herein for rotor speed control. The new controller featuring with a single-neuron network is able to adaptively tune the gains (weights) online. The simple structure of the controller reduces the computational load and facilitates the algorithm implementation on low-cost hardware. Finally, the proposed controller is validated by numerical simulations and experiments on the MTE in laboratory conditions, and the results show that the proposed controller achieves remarkable effectiveness for speed tracking control. In comparison with the PID controller, the proposed controller yields 54% and 66% reductions on static tracking error under two typical cases.

## 1. Introduction

The turbojet engine is recognized as a powerful candidate propulsion system for unmanned aerial vehicles (UAVs) [1]. In the past decade, owing to the extremely high thrust-to-weight ratio, small-sized turbojet engines, especially micro-turbojet engines, are becoming attractive for the application in UAVs or remotely-controlled airplanes.

The instability induced by unsteady fluids poses a challenge in turbojet engine design and operation for UAVs. The sharp change in fuel flow may induce a compressor surge (over-temperature) or flame-out. In practice, some stability boundaries like the surge lines are usually given in the compressor map to limit the operation range of turbojet engines [2]. On the other hand, turbojet engines are usually expected to have a quick response to command from UAVs. The fast acceleration or deceleration of engines easily results in the undesired surge or flame-out. Thus, it is necessary for turbojet engines to be equipped with an effective controller to guarantee reliable operation. 

Generally, the purpose of the engine control system is to ensure the safety and reliability, affordability and capability of the propulsion system. Specifically, it optimizes the engine transient response for accelerating and decelerating smoothly, quickly and correctly. It also prevents the possible surge/stall, overpressure and over-temperature. Besides, reduction of fuel consumption [3,4,5], fault diagnosis [6,7], vibration control [8], and fault-tolerant control [9] are also significant for engine control systems of UAVs. Moreover, the engine control system should have strong adaptivity when the system parameters vary. The fuel flow rate and the nozzle area changes are the most important manipulated input variables [10]. Since the engine thrust is proportional to the rotor speed of the compressor, the rotor speed is usually measured and practically used as an output variable for closed-loop feedback [11]. The micro-turbojet engines have a limited nozzle area, and consequently, the control system can be simplified as a single-input (fuel flow rate) and single-output (rotor speed) system. 

Like most industrial automatic control, the proportional–integral–derivative (PID) controller has been recognized as an efficient method for engine control in UAVs because of its simple structure and high reliability [12]. However, the PID method is not the ideal strategy because it may deteriorate the performance when dealing with time-varying and nonlinear systems. Therefore, various new variants of the PID algorithm have been intensively investigated in an attempt to improve the PID controller performance. A proportional-integral fuzzy logic controller was proposed in [13] for the thrust regulation of small scale turbojet engines. Finally, the controller was tested on the Pegasus MK3 micro-turbine, and the experimental results confirmed that the controller had quick response and high tracking precision. In [14], a nonlinear fuzzy logic-based PID controller was developed by incorporating the fuzzy adaptive Unscented Kalman Filter into the control system. Simulation results revealed that the new approach yielded superior performances over the traditional PID controller and was capable of reducing the distortions induced by noises. Some other advanced engine control strategies have been developed for performance enhancement of the engine system. In [11], the genetic algorithms are employed to optimize the parameters in the fuzzy logic controller so that the controller can meet the performance requirements and safety constraints. In [15], the fuzzy norms are used to improve Min–Max controller performance for turbojet engine. In addition, the linear model is considered in [16] and stability analysis is performed for a Min–Max multi-regulator system. In [17], an H infinity controller is designed for an identified model of gas turbine GE9001E so as to maintain the speed and the exhaust temperature within desired range. In [18], the H infinity controller is deeply investigated for an advanced turbofan engine, but the engine is also described by a linear model. The linear quadratic integral (LQI) algorithm is used to design a robust control system in [10], and the variable exhaust nozzle and fuel mass flow rate acting as the inputs result in a multiple-input multiple-output system. Since the predictive properties and physical constraints can be directly included in the control framework, a model predictive controller is employed for the turbine engine system in [19], and the fuel and air flow are considered as physical constraints. The rate-based model for MPC design is introduced in [20], and errors between linear model predictions and the response of the actual nonlinear system is compensated by extra offset states. However, it is worth noting that the conclusions of above methods are mostly drawn from numerical simulations and have not been verified by any experimental test. Besides the above-mentioned strategy study, a new control framework called intelligent situational control is presented for integration of the control system with a diagnostic system [21]. This new approach was validated in real-world laboratory conditions using a unique design of small turbojet engine iSTC-21V. 

Most subsystems associated with UAV, as well as the turbojet engine, can be viewed as nonlinear systems with strong variability [22,23]. The neural network (NN) has been recognized as an efficient means for nonlinear time-varying controller design because of its learning ability. Numerous researchers have put a lot of effort into the application of NN to UAV [24,25,26]. But it should be noted that the NN featuring with multiple hidden layers and neurons puts forward high demands on the computational ability and memory resources. Low-cost hardware like the STM32 microcontroller has limited computational ability and memory resources because of its low price. For example, the STM32F103C8T6 only costs two dollars and has only 64 kb RAM and 512 kb ROM, which can hardly meet the requirements of the NN models. Therefore, it is difficult to realize the NN on low-cost hardware such as a microcontroller. For this reason, some NN-based PID schemes or PID-like NNs with self-tuning capability have been developed because of their simplicity and easy implementation [27,28,29]. However, they have not (to the authors’ knowledge) previously been applied to the turbojet engine studied here, and no successful application has been reported on a real turbojet engine.

In this article, we design and realize a full authority digital electronic controller (FADEC) based on the single neural adaptive PID (SNA-PID) strategy for NT-50A, a micro-turbojet engine (MTE) developed by the authors. The aim of this work is to explore a robust and adaptive control system with online adjustment for MTE. Differing from the traditional PID controller with fixed parameters, the SNA-PID developed herein can adjust the weights adaptively online and guarantee the engine’s adaptive capacity and robustness. On the other hand, the simple implementation of the SNA-PID controller allows us to implement the algorithm on low-cost hardware like the STM32 microcontroller. Finally, the validating experiment demonstrates that the proposed control system is able to yield remarkable speed tracking performance. 

The remaining part of the paper proceeds as follows. Section 2 begins with the description of the turbojet engine system for experiments and the objectives of this study. Then, the control system and the control algorithm based on SNA-PID are introduced in Section 3. Section 4 presents the simulation results based on the SNA-PID controller. Different experiments were undertaken to evaluate the performance of the SNA-PID controller and experimental results are presented in Section 5. Finally, Section 6 briefly concludes the paper by summarizing the main results.

## 2. Problem Description

### 2.1. Description of Micro-Turbojet Engine (MTE) System

Turbojet engines are the most commonly used power generator in modern aircraft. As shown in Figure 1, a turbojet engine is mainly composed of five parts: (1) Inlet, (2) compressor, (3) combustion chamber, (4) turbine, and (5) nozzle. The air is ingested through the inlet, in which the airspeed is controlled within a proper range. The compressors consisting of blades compress the air to increase its pressure and temperature. The high-pressure air then enters the combustion chamber in which the fuel nozzle emits fuel that is mixed with air and ignited. The high-pressure gas is transferred through the turbine where it rotates the turbine. It then erupts backward through the nozzle. Since the turbine is mounted on the same shaft as the compressor, it also rotates the compressor. Consequently, the air is compressed and the turbine is rotated repeatedly. An auxiliary electric start motor is generally used for starting and cooling the jet engine.

A micro-turbojet engine NT-50A developed by Northwestern Polytechnical University is used as the platform for experimental testing and verification of the proposed controller. The photo of the engine is shown in Figure 2. Some basic engine specifications measured under the laboratory condition are listed in Table 1. 

For ease of control, the KMZ10CM speed sensor and K-Thermocouple are employed here for measurements. The KMZ10CM is an extremely sensitive magnetic field sensor that employs the magnetoresistive effect of the thin-film permanent magnet material. It can generate a signal proportional to the intensity or direction of the magnetic field. The KMZ10CM sensor works with a magnet ring mounted on the bearing of the engine that is close to the inlet (as shown in Figure 3). Once the magnet ring rotates with the bearing, the corresponding magnetic field will vary periodically. Consequently, the sensor outputs a sinusoidal signal, achieving the non-contact speed measurement. In addition, a signal processing circuit consisting of an operational amplifier (LM358) and voltage comparator (LM393) is designed to amplify the weak signal and convert the sinusoidal signal into the equal-period pulse signal. The rising edge interval is then captured by the electric controller unit (ECU) so as to obtain the revolutions per minute.

In order to measure the exhaust temperature, the K-Thermocouple is placed at the rear of the turbine. The measured analog voltage signal from the temperature sensor is digitalized by MAX6675 later, which provides the cold-junction compensation and acquires the data with the 12-bit revolution. The measurement accuracy can be up to 0.25 °C and a wide range of 0–1024 °C is allowed.

The fuel pump (ZP25M14F, HP-Tech, Austria) is the most important actuator in an engine control system, which is driven by a DC motor in charge of regulating fuel flow injected into the combustion chamber. However, it is hard to measure the fuel flow rate accurately because of its very small magnitude. Hence, there is no sensor specially designed to sense the fuel flow rate in our work. We regulate the flow rate by changing the duty cycle of pulse width modulation (PWM) pulses driving the DC motor.

For the easy understanding of the control requirements, the whole operation of the micro-turbojet engine can be systematically decomposed into the following frames:Start-up control. The engine start-up process can be divided into six parts in our work: (1) Burner on. After the starting operation, the igniter is switched on for preheating, which takes 10 s to fully preheat. (2) Starter on. The starter motor is turned on to rotate the rotor to obtain a lower speed state, which lasts 10 s. (3) Ignition. The gas valve is opened to supply gas or tiny bits of fuel, and the temperature of the combustion chamber rises gradually. A successful ignition is marked by the combustion chamber temperature rising to 90 °C. (4) Preheat. The gas valve and fuel valve are opened alternately, and the fuel supply is increased gradually. When the combustion chamber temperature rises to 150 °C, preheating is completed. (5) Switch-over. The gas valve is closed and the fuel valve is open continuously in this process, so that the engine enters the stable fuel supply stage. (6) Fuel ramp. The fuel supply is increased and the starting motor accelerates so that the engine enters the acceleration process. The starting motor will be shut down when the engine speed reaches the idle speed. During the start-up process, only some logic controls are required. The engine works under the open-loop conditions with limitations on the fuel flow rate.Operational control. The ECU receives the target speed command from the flight control system of UAV and utilizes it as a reference value of the closed-loop control system. The tracking control is required to ensure the fastest possible acceleration or deceleration of the engine without exceeding its operational envelope. Once the engine reaches the desired speed, the tracking control will prevent the constant speed from fluctuation.Shutdown control. This involves the disabling fuel flow supply and closing of electromagnetic fuel valves.

### 2.2. Objectives of This Study

The topic of this paper mainly focuses on operational control. Although the closed-loop system can be simplified as a single-input (fuel flow rate) single-output (rotor speed) system, we still face significant challenges in the MTE controller design.

The turbojet engine is a highly nonlinear system. At low speeds especially, the MTE shows strong nonlinearity because of very small fuel consumption. A single linear controller like PID controller designs cannot guarantee superior performance over the entire operating range. Gain scheduling is a practical approach for nonlinear systems control using a family of linear controllers, each providing satisfactory control for a different operating point of the system. But such a complex solution heavily relies on model sets and pre-knowledge. The model-free adaptive control is very promising for MTE.The sensors for measurement play dominant roles in closed-loop feedback control. The sensor for speed measurement is very sensitive to external disturbance (e.g., the engine vibration). The resulted measurement noise seriously deteriorates the tracking control performance. Additionally, the absence of adequate microflow meter also poses a challenge to accurate flow rate regulation.Due to the limited hardware resource of the embedded system, the complicated but advanced algorithm cannot be implemented online with low-cost controllers. PID control strategy is widely used in industrial applications due to its simplicity and efficiency. However, its performance degrades significantly when the system state deviates from the equilibrium point at which the PID is tuned. Therefore, a simple adaptive algorithm that can be implemented on low-cost hardware like a microcontroller is urgently demanded.

## 3. Control System Realization for MTE

### 3.1. Control System Design and Realization

The basic architecture of the control system designed for MTE is shown in Figure 4. The electronic control unit (ECU) plays a central role in the control system. The core of the ECU is a 32-bit microprocessor in charge of control algorithm implementation. The ECU receives the target speed command delivered from the UAV flight control system and utilizes it as a reference value in the control system. Simultaneously, the ECU collects speed and temperature measurements from the speed sensor and K-Thermocouple respectively and converts the analog measurements into digital signals for feedback control during the engine operation. With the speed and temperature information, the ECU generates various control signals for switching magnetic valves or regulating the fuel flow rate. The fuel pump driven by a DC motor is used as an actuator to regulate fuel flow injected into the combustion chamber. In practical realization, the PWM signal with a varying duty cycle, taken as the control signal, is modulated by the ECU and delivered to the pump motor.

An important consideration in the MTE control system is to ensure the engine does not surge and flame out during acceleration or deceleration. As shown in Figure 5, the MTE should be operated in a limited safe range. The maximum fuel limit protects against a surge and stall, over-temperature, over speed, and overpressure. The minimum fuel limit protects against combustor blowout. In addition, the change rate of the fuel flow has to be limited in acceleration and deceleration boundary. Thus, the engine control should consider the limitation on input.

In our control system, the temperature sensor is mainly used for temperature monitoring and indicator of sequence logical control. Its functions can be described in three stages: (1) Start-up. Determines whether the combustion chamber is ignited. When the temperature reaches the setting point, fuel valves are switched to accelerate the engine to idle; (2) Operation. Determines whether the over-temperature occurs during normal operation to prevent engine damage; (3) Shutdown. Determines whether the engine is completely cooled and if it is necessary to run the starting motor to blow the air for cooling the combustor. It is clear that the accurate and continuous regulation of fuel flow rate heavily depends on the rotor speed that is proportional to generated thrust, while the temperature sensor mostly serves as an indicator for logical control. Therefore, the main content of MTE control is accurate speed tracking control.

We design the ECU and built a prototype shown in Figure 6. The ECU consists of a 32-bit microprocessor (STM32F103C8T6), a driver circuit for the fuel pump and fuel valves, a sensors circuit for temperature and speed measurement, and power supply circuitry. The ECU communicates with an external system like the flight control system via the RS-232 serial interface. It receives command from the external system and reports the statuses of the engine. The ECU automatically regulates the engine speed by adjusting the driving signal of the fuel pump. It also performs the tests of actuators and sensors, fault detection, data recording, and reporting. When the engine fails, the program will automatically restore the relevant parameters in FLASH and restart the engine again. Note that all parameters involved in the engine operation control are configurable in software.

### 3.2. Control Algorithm Based on SNA-PID 

To cope with the nonlinearity and sensitivity, an adaptive single neural adaptive PID algorithm is developed here for turbojet engine control, and is performed on the aforementioned ECU. The self-tuning capability ensures perfect tracking performance throughout the entire operating range. 

In order to derive the relationship needed to design SNA-PID controller, classical neural network theory is used [26,27,28]. First the traditional PID controller is introduced; it has the advantage of a simple framework and easy implementation, and it is closely linked to expected performance. The discrete PID expression is written as:(1)$$u(k)=K_{p}e(k)+K_{i}T\sum_{j=0}^{K}e(j)+\frac{K_{d}}{T}[e(k)-e(k-1)]$$

The corresponding incremental PID controller can be written as:(2)$$\Delta u(k)=K_{p}[e(k)-e(k-1)]+K_{i}Te(k)+\frac{K_{d}}{T}[e(k)-2e(k-1)+e(k-2)]$$
where e(k) is the error between the desired output and actual output, and $K_{p}$
$K_{i}$ and $K_{d}$ are respectively the proportion, integration and differential coefficients.

However, the PID controller with fixed parameters cannot meet the requirements of high-performance control when the operation condition changes. In this section, a combination of a single neuron and an incremental PID controller, called the SNA-PID controller, is presented to overcome the limitation of parameter tuning in the nonlinear systems. The control interconnection of the MTE system with the SNA-PID controller is depicted in Figure 7. It is not only simple in structure, but it is also adaptable to environmental changes, and has strong robustness.

According to the above structure, r(k) and y(k) are the reference input (desired speed) and actual output (measured speed) of the system, respectively, and the error satisfies e(k)=r(k)−y(k). It is assumed that y(k) is measurable. In the view of the SNA-PID controller, the input is e(k) and the output is the incremental duty cycle of PWM ∆*u*(*k*) driving the pump motor. Similar to a normal PID controller, the variables $x_{1}$, $x_{2}$ and $x_{3}$ satisfy the following relations respectively: (3)$$\begin{array}{l} x_{1}=\Delta x_{2}=e(k)-e(k-1)\\ x_{2}=e(k)\\ x_{3}=\Delta x_{1}=e(k)-2e(k-1)+e(k-2) \end{array}$$

The control signal *u*(*k*) at *k*th sampling time is then obtained through:(4)u(k)=u(k−1)+Δu(k)
with
(5)$$\Delta u(k)=K[w_{1}(k)x_{1}(k)+w_{2}(k)x_{2}(k)+w_{3}(k)x_{3}(k)]$$
*K* is the proportional coefficient of the single neuron and K>0. $w_{1}$, $w_{2}$ and $w_{3}$ are respectively the weight values of $x_{1}$, $x_{2}$ and $x_{3}$.

The SNA-PID controller aims at minimizing the error between the reference value and the measurement of the closed-loop system. Considering the well-known mean square error (MSE), the cost function *J*(*k*) at sampling times *k* + 1 is defined as:(6)$$J(k+1)=\frac{1}{2}{\left(r(k+1)-y(k+1)\right)}^{2}=\frac{1}{2}{\left(e(k+1)\right)}^{2}$$

If the method of gradient descent is adopted here to minimize the cost function *J*(*k*), then the gradient descent can be formulated as:(7)$$w_{j}(k)=w_{j}(k-1)-\eta_{j}\nabla w_{j}J(k)$$
where the $\nabla w_{j}J(k)$ is the gradient vector of *J*(*k*). It implies the direction of weight updating is along the negative gradient direction. $\eta_{j}$ is the learning rate of the hidden layer. Thus, applying the chain rule, the rule of updating weight is written as:(8)$$\begin{array}{ll} \Delta \;w_{j}\left(k\right) & =-\eta_{j}\nabla w_{j}J(k)=-\eta_{j}\frac{\partial J(k)}{\partial w_{j}(k)}=-\eta_{j}\frac{\partial J(k)}{\partial y(k)}\frac{\partial y(k)}{\partial u(k)}\frac{\partial u(k)}{\partial w_{j}(k)}\\ & =\eta_{j}Ke(k)x_{j}(k)\frac{\partial y(k)}{\partial u(k)} \end{array}$$

It should be noted that the model of MTE between the output y(k) and the input u(k) is not pre-known. Theoretically, ∂y(k)/∂u(k) can be approximately estimated by Δy(k)/Δu(k), but the speed measurement noise will lead to a large estimation error. As suggested in [30], the term ∂y(k)/∂u(k) may be approximated using the gradient sign rather than its actual value, i.e., the expression for ∂y(k)/∂u(k) becomes $\mathrm{sgn}\left(\frac{\partial y\left(k\right)}{\partial u\left(k\right)}\right)$. 

If we revisit Figure 5, it is easy to find that the speed in steady-state is a monotone increasing function of steady fuel flow rate. Furthermore, since the fuel flow rate is proportionally regulated by the duty cycle of PWM pulses (*u*(*k*)), then $\frac{\partial y\left(k\right)}{\partial u\left(k\right)}>0\;\mathrm{and}\;sgn\left(\frac{\partial y\left(k\right)}{\partial u\left(k\right)}\right)=1$. Differing from previous work, the value of ∂y(k)/∂u(k) is represented here by a positive value β(k), instead of $sgn\left(\frac{\partial y\left(k\right)}{\partial u\left(k\right)}\right)$. For simplicity, Equation (8) can be rewritten as
(9)$$\Delta w_{j}(k)=\eta_{j}Ke(k)x_{1}(k)\beta (k)={\eta^{\prime }}_{j}e(k)x_{j}(k)\beta (k)$$

Theoretically, with sufficient pre-knowledge of the model, the varying value of β(k) can be pre-determined from Figure 5. In our work, a model-free solution is proposed by replacing β(k) with input u(k) (the duty cycle). At the low speed, the ∂y(k)/∂u(k) is relatively small. With the growth of speed, ∂y(k)/∂u(k) gradually increases. Such variation trend completely coincides with the change of u(k).
(10)$$w_{j}(k+1)=w_{j}(k)+{\eta^{\prime }}_{j}e(k)x_{j}(k)u(k)$$

The replacement leads to a non-steepest descent, but it simplifies the algorithm and guarantees a correct minimum search direction. A similar expression can also be found in references [31], but no reasonable explanation is given in that work. 

In practice, in order to ensure the convergence and robustness of SNA-PID, the weights are required to be normalized before calculation of the control variable. Therefore, Equation (5) can be rewritten as: (11)$$\Delta u(k)=K\sum_{i=1}^{3}{w^{\prime }}_{i}(k)x_{i}(k)$$
and
(12)$${w^{\prime }}_{i}(k)=w_{i}(k)/\sum_{i=1}^{3}\left|w_{i}(k)\right|$$
with
(13)$$\begin{array}{l} w_{1}(k+1)=w_{1}(k)+{\eta^{\prime }}_{1}e(k)u(k)x_{1}(k)\\ w_{2}(k+1)=w_{2}(k)+{\eta^{\prime }}_{2}e(k)u(k)x_{2}(k)\\ w_{3}(k+1)=w_{3}(k)+{\eta^{\prime }}_{3}e(k)u(k)x_{3}(k) \end{array}$$
where ${\eta^{\prime }}_{1}$, ${\eta^{\prime }}_{2}$ and ${\eta^{\prime }}_{3}$ are respectively proportion, integration and differentiation learning rate. 

On the other hand, fast acceleration and deceleration may induce a sudden change in the fuel flow rate, when Δu(k) totally depends on the controller. Once the large increment or decrement exceeds the physical limits of the engine, over-temperature, over-speed or flameout will happen. Therefore, it is necessary to set limits on the output of the controller to guarantee the stable operation of the engine. The limitation of the Δu(k) over different speeds is defined as
(14)$$\Delta u(k)=\{\begin{array}{cc} \begin{array}{l} \Delta u_{ac}\\ \Delta u(k)\\ -\Delta u_{dc} \end{array} & \begin{array}{l} \Delta u(k)\leq \Delta u_{ac}\\ -\Delta u_{dc}<\Delta u(k)<\Delta u_{ac}\\ \Delta u(k)\leq -\Delta u_{dc} \end{array} \end{array}$$
where $\Delta u_{ac}$ and $-\Delta u_{dc}$ are the upper and lower boundary of Δu(k) respectively. They pose constrains on MTE acceleration and deceleration.

In fact, the flow rate *u*(*k*) corresponding to a specific speed should have the corresponding boundary. The real limit values are mainly determined by the experiments in an open-loop system. The stepped increase and decrease will help us gradually explore the limits. According to our experimental experience, the limits are seriously affected by the following factors:Fuel Pump. Under the same driving signal, the fuel supply capacity varies with fuel pumps in practice.Type of fuel. MTE can be compatible with many different fuels, such as aviation kerosene and diesel. Different types of fuel have different combustion characteristics that have a great influence on speed control in the MTE system.The operating conditions. At low speeds, the engine is sensitive to fuel fluctuations, and the limit value will be smaller. On the contrary, MTE at high speed is no longer sensitive to fluctuations due to the increased fuel consumption of the engine, and the increment limits can be appropriately relaxed.

## 4. Modeling and Numerical Simulation

### 4.1. Model Identification

A dynamic mathematical model of the turbojet engine is indispensable to numerical simulation. This section aims at establishing mathematical models for MTE under real operating conditions. Differing from the theoretical modeling method, the system identification technique is employed here to obtain the model from experimental data. Taking into account the nonlinearity of MTE, a set of linear models corresponding to different operating conditions are identified from experimental data.

As mentioned in [32], the experimental identification is performed by applying the PWM pulses of 10 kHz with varying duty cycles to the fuel pump. Meanwhile, the revolutions per minute (RPM) of the engine is recorded by the designed ECU. The input signal of the identification is the duty cycle in the continuous step waveform. All steps have the same amplitude, and the corresponding step response is the measured RPM. Figure 8 shows the input duty cycle and rotation speed response of the engine without any control strategy.

It is worth noting that MTE is a time-varying system, but the dynamic behavior of MTE at specific operation points (specific speed) can be represented as a linear model. The second-order transfer function can be employed to define the structure of individual linear models by: (15)$$G(s)=\frac{b_{0}}{a_{2}s^{2}+a_{1}s+a_{0}}$$
where $a_{i}$, $b_{i}$ are the constant coefficients. The structure of Equation (15) was selected according to the experimental results in [33].

For this reason, within the entire operating speed range, the dynamic behavior of MTE can be approximately described by a set of second-order models. Each model can be identified by using step input and step response measurements corresponding to a specific speed range. We employ the *tfest* function in MATLAB to estimate the transfer function in Equation (15). Table 2 shows the identified parameters of the engine within the operating range from 51,900 to 70,800 RPM.

The parameters varying with speed further proves the time-varying property of MTE. To validate the identified models, the real input is also applied to the estimated transfer functions. The calculated and measured responses are compared in Figure 9. In order to confirm the accuracy of the estimated linear model, quality metric is employed to evaluate the fitting of the transfer functions obtained in fixed operational points. Specifically, normalized root mean squared error (NRMSE) is expressed as a percentage and represents the fitting percent between the estimated function and the true experimental data, defined as:(16)$$FittingPersent=(1-\frac{\left\|y_{measured}-y_{\mathrm{mod}el}\right\|}{\left\|y_{measured}-\bar{y_{measured}}\right\|})\times 100\%$$
where $y_{measured}$ is the measured output data, $\bar{y_{measured}}$ is the mean of $y_{measured}$, $y_{model}$ is the simulated response of the model and ‖.‖ indicates the 2-norm of a vector. The specific fitness percentage corresponding to fixed points is shown in Table 2. According to the numerical results and Figure 9, the simulation results based on the estimated models can approximate the measured data with good fitting percent. Therefore, the linear models in Table 2 can be used for the representation of engine working at specific points.

### 4.2. Numerical Simulation and Results

Several simulations are carried out in this section to evaluate the performance of the SNA-PID controller. The six identified linear models are used here to simulate the open-loop dynamics of MTE. A comparison between the SNA-PID controller and the conventional PID controller has been made. For a fair comparison, the initial settings of weights in the SNA-PID controller are totally the same as the coefficients in the PID controller. More specifically, for the PID controller, *K_p_* = 0.1989, *K_i_* = 0.2980, *K_d_* = 0.0004; for the SNA-PID controller, *K* = 0.2, and the initial values of weights in Equation (10) are $w_{1}\left(0\right)=0.9946,w_{2}\left(0\right)=0.0298,w_{3}\left(0\right)=0.0995$ respectively. The learning rates in Equation (13) are tuned to $\eta_{1}^{\prime }=2.5\times 10^{-5},\;\eta_{2}^{\prime }=6.25\times 10^{-6},\eta_{3}^{\prime }=6.25\times 10^{-6}.$ The sampling time interval of the control system is 0.02 s.

The engine is simulated to gradually increase the rotor speed from 51,900 to 70,800 RPM. The tracking performances of the two controllers within the speed range are compared and depicted in Figure 10. The larger overshoot and longer setting time indicates that the SNA-PID controller has worse performance than the PID controller at the low-speed range. However, as the number of iterations increases, the SNA-PID controller shows great adaptivity. At the medium-speed range, the two controllers yield almost the same performance. With higher speed, the SNA-PID controller has a great advantage over the PID controller.

To gain a deep insight into the performance evaluation, some indicators closely related to performance are reported in Table 3. It is clear that the SNA-PID controller has a longer rising time and setting time than the conventional PID controller at the first speed range 51,900–55,500 RPM. The overshoot of the SNA-PID controller is 6.67%, which is also obviously larger than the 2.78% of the PID controller. These indicators show that the performance of the SNA-PID controller at the beginning of iterations is worse than that of the PID controller. However, as the number of iterations increases, the SNA-PID controller shows great self-tuning ability. At the medium-speed range (58,900–61,900 RPM), the SNA-PID controller has yielded a comparable performance to the PID controller. At the high-speed range (67,800–70,800 RPM), the performance of the SNA-PID controller completely surpasses that of the PID controller. It is easy to find that all the indicators related to SNA-PID gradually decrease with the growth of rotation speed. The rising time decreases from 1.330 s to 1.155 s, and the setting time declines from 3.051 s to 2.361 s simultaneously. Of particular note, the overshoot is eliminated. On the contrary, there is no obvious improvement in the PID controller. These results suggest the strong self-tuning capability of the SNA-PID controller.

## 5. Experiments and Discussion

### 5.1. Experimental Setup

Experiments are carried out for control performance evaluation. The experimental setup is shown in Figure 11. The MTE named NT-50A is employed in our experiment with a fixed exhaust nozzle. The MTE is fixed on the test bench and is connected with ECU by signal and power lines. The electric fuel pump (ZP25M14F, HP-Tech, GmbH, Maria Buch-Feistritz Austria) absorbs light diesel oil from the fuel tank and provides it to the engine. The environmental temperature is 35 °C.

The ECU equipped with a microprocessor (STM32F103C8T6, STMicroelectronics) works as the intermediate managing the communication between the host computer and MTE, as well as the controller. STM32 is a 32-bit microcontroller based on ARM core. It has rich ports for easy connections with diverse sensors. Due to the high performance and low cost, STM32 is one of the essential controllers in the industry field. We use STM32 herein as the main control chip for algorithm implementation. The temperature and rotor speed are acquired by the internal ADC of microprocessors with a sampling frequency of 50 Hz. Meanwhile, the ECU sends the sampling data to the host PC via a serial port. For ease of communication, a user interface based on LabWindows/CVI platform has been developed on the host PC. The designed user interface is shown in Figure 12. It is used to send reference speed commands to the ECU and to receive the engine status from the ECU. In this way, the engine operation can be monitored by on-screen displays, and the operation of the engine can be timely stopped in urgent circumstances. In our experiments, the interface helps us record the testing data to hard disk synchronously for subsequent analysis.

### 5.2. Experiments and Results

In order to validate the effectiveness and superiority of the proposed SNA-PID controller, we perform comparative experiments between SNA-PID and normal PID. In order to gain a full understanding of control performance under different operating conditions, the experiments are carried out within the operating range of 50,000 RPM to 100,000 RPM. Similar to the simulations, the tracking performances under different operation speeds are investigated. The experiments start from the idle speed (50,000 RPM), at which speed the feedback control starts working. 

It is worth noting that limitations on the Δu(k) described in Equation (14) have great impacts on control performance. For this reason, two sets of limits are considered and compared in our experiments. In the first set (Case I), the Δu(k) is limited in a narrow range, while the Δu(k) in the second set (Case II) exists in a wider range. The specific values corresponding to the two sets of limits are listed in Table 4. In practice, the limits can be determined by the continuous increment (decrement) of duty cycle of the PWM signal. Once the abnormal states occur, such as over-temperature, flameout and so on, the increment (decrement) is stopped and the current increment (decrement) is recorded as the limit value. In addition, according to our experimental experience, the limits are seriously affected by the following factors: Fuel pump, type of fuel and the operating conditions. 

The well-tuned PID controller and SNA-PID controller are used for comparison. The parameters in the PID controller with incremental PID algorithm are set as follows:$$K_{p}=0.04;K_{i}=0.11;K_{d}=0$$
which were tuned according to experience. 

For the SNA-PID controller, *K* = 2, the initial values of weights in Equation (10) are
$$w_{1}(0)=1.1,w_{2}(0)=0.04,w_{3}(0)=0.01$$

The learning rates are set as
$$\eta_{1}^{’}=8.5\times 10^{-4},\;\eta_{2}^{’}=1.4\times 10^{-4},\eta_{3}^{’}=1.0\times 10^{-5}$$

For ease of control performance evaluation, the step responses of the MTE under control are compared in Figure 13. It is clear that the two controllers in two cases show good performance on speed tracking. Due to the strong nonlinearity of MTE at low speed (50,000–60,000 RPM), the obvious overshoots are observed in two cases. In comparison with the PID controller, the SNA-PID controller shows better tracking performance. It yields smaller overshoots and faster decay. Furthermore, with the growth of speed, the benefit of the SNA-PID controller is becoming increasingly apparent. The rising time as well as the overshoot are gradually reduced by the SNA-PID controller. Besides, it is worth noting that the SNA-PID controller also brings a great reduction in static error. This implies that the proposed control system is able to provide a more stable thrust force than the traditional system at the desired operating speed. 

To quantify the tracking performance of two controllers, we list the rising time, setting time and overshoot corresponding to two cases in Table 5 and Table 6 respectively. It is clear that the SNA-PID controller shows great self-learning ability, and the performance enhancement has been validated by continuous decreases in rising time and setting time. As illustrated in Table 5, the rising time and setting time in Case I is gradually decreased by the SNA-PID controller from 1.4 s to 0.63 s, and 2.2 s to 0.67 s respectively. On the other hand, the PID controller with fixed parameters has no adaptivity. It only shows the superior performance at a specific speed range (like 60,000–70,000 RPM), since the parameters of the controller are specially tuned to a given operating point. It is worth noting that the PID controller achieves less overshoot at the expense of more rising time and setting time. On the contrary, the SNA-PID controller is capable of a yield comparable overshoot with less rising time and setting time. Therefore, it can be concluded that the SNA-PID controller is capable of generating a quicker response with fewer speed fluctuations than the PID controller.

The different limitations on the Δu(k) also impacts the performance. From Figure 13b, we found that the relaxed limitations in Case II helped the MTE reach the desired speed (60,000 RPM) with less time, but it also induced obvious overshoot. To gain a deep insight into the comparison, we revisit Table 5 and Table 6. The relaxed limitations are obviously beneficial to the performance enhancement of the SNA-PID controller. Compared with Table 5, the reductions of rising time and setting time are easily found in Table 6. Although the relaxed limitation induces a larger overshoot at the low speed, the SNA-PID controller shows sufficient ability to eliminate the overshoots with an increase of iterations. As shown in Table 6, the overshoot at the highest speed (90,000–100,000 RPM) turned out to be zero. In contrast to the SNA-PID controller, the relaxed limitation is of little significance to the PID controller. The only obvious improvement in rising time is observed at the low-speed range. There is no clear enhancement of other indicators at high- or medium-speed range.

Besides the above-mentioned indicators, the modified mean square error (MSE) is also employed here to evaluate the tracking performance. It is of the form
(17)$$MSE=\frac{1}{10000N}\sum_{i=1}^{N}{\left(n_{refer}(i)-n_{measure}(i)\right)}^{2}$$
where *N* is the length of the data, $n_{set}$ and $n_{mea}$ are the desired speed and the measured speed respectively. In addition, the measures of holistic tracking and static tracking performance are notated as $MSE_{h}$ and $MSE_{s}$ respectively. The difference between them is that only data after setting time in each speed range are selected for calculation of $MSE_{s}$, while $MSE_{h}$ are obtained from all recorded data. 

Figure 14 shows that the track errors generated by the SNA-PID controller are much less than that generated by conventional PID. Especially, the SNA-PID controller demonstrates a great advantage in static tracking performance. In comparison to the PID controller, the proposed controller yields 54% and 66% reductions on static tracking error under two typical cases. All the comparative results further evidence the superiority of the SNA-PID controller. Meanwhile, Figure 14 also illustrates that the relaxed limitations in Case II bring a reduction in MSE. Therefore, the relaxation of limitation is clearly beneficial to performance enhancement. 

For easy understanding of the self-learning ability of the SNA-PID controller, the recorded time histories of three weights in Equation (13) are plotted in Figure 15. All the weights adapt to the variation of the desired speed and have great changes at each instant of speed transition. Once the actual speed reaches the desired value, the weights of the SNA-PID controller will stop the update. On the other side, Figure 15 also illustrates that the limitations on Δu(k) have great impacts on the weights, especially at the low-speed range. In comparison with Case II, the limits in Case I pose stricter constraints on flow rate regulation, thus larger variations of weights are generated by the SNA-PID controller to remedy the control deficiency. Unfortunately, the larger variations may induce larger tracking error, particularly static error as demonstrated in Figure 14. On the contrary, the relaxed limits in Case II provide sufficient control capability. It allows the controller to update the weights with the smaller variations. It is worth noting that the variations of weights in both cases get smaller and smaller because of the adaptive regulation of the SNA-PID controller. 

## 6. Conclusions

In this article, the SNA-PID controller for MTE speed control is proposed and experimentally validated. The performance of the proposed controller is evaluated in numerical simulation and applied to the real operation of an MTE. In order to confirm the superiority and effectiveness, a comparison between the proposed controller and conventional PID technique has been made in two typical cases. All the selected indicators, ranging from rising time, setting time and overshoot to MSE, demonstrate that the SNA-PID controller with self-tuning ability is capable of effectively tracking the speed command, and yields a better performance than the PID controller.

## Figures and Tables

**Figure 1 sensors-20-00345-f001:**
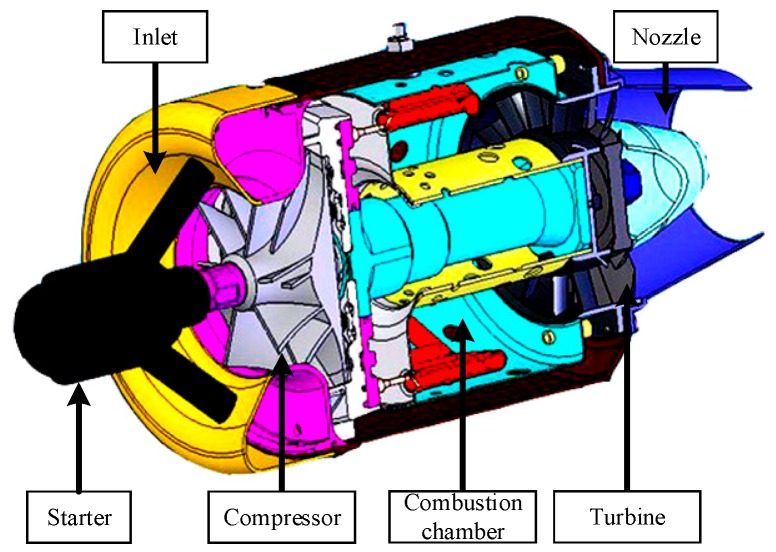
Turbojet engine structure.

**Figure 2 sensors-20-00345-f002:**
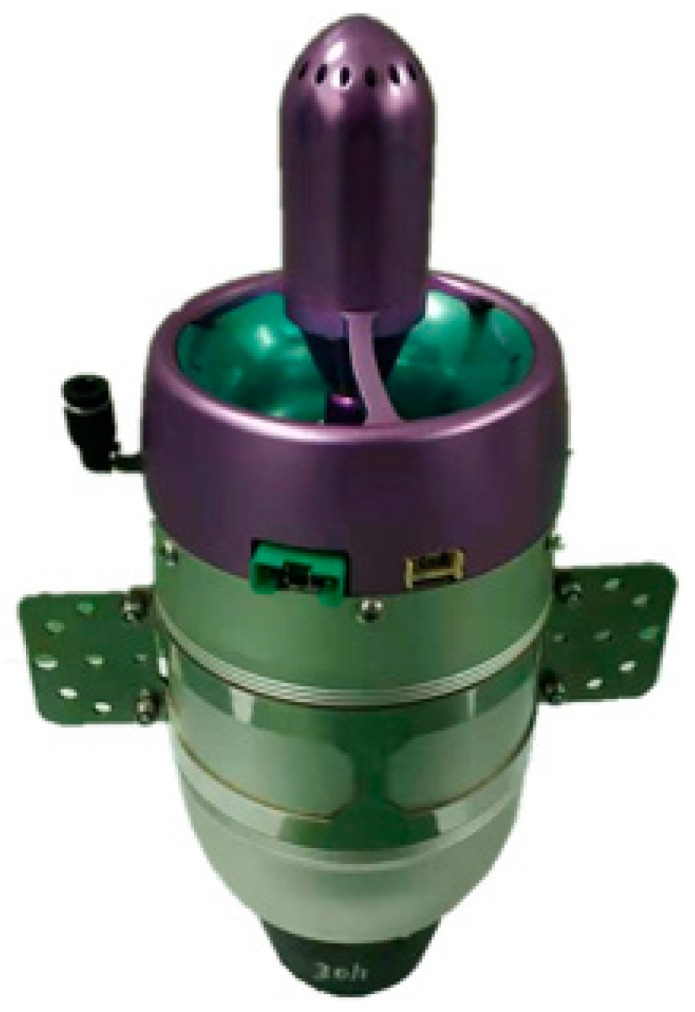
NT-50A micro-turbojet engine (MTE).

**Figure 3 sensors-20-00345-f003:**
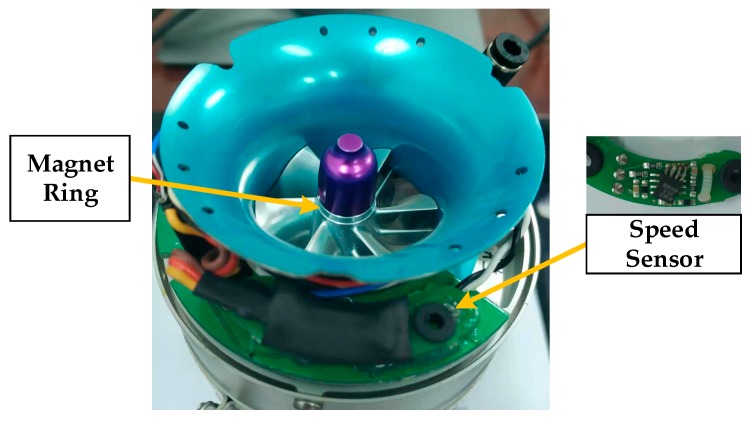
The installation of speed sensor KMZ10CM.

**Figure 4 sensors-20-00345-f004:**
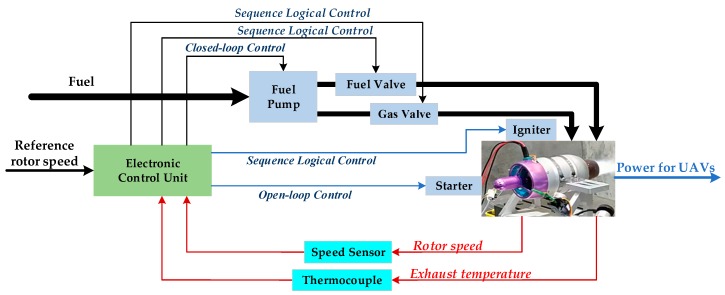
The control system for MTE.

**Figure 5 sensors-20-00345-f005:**
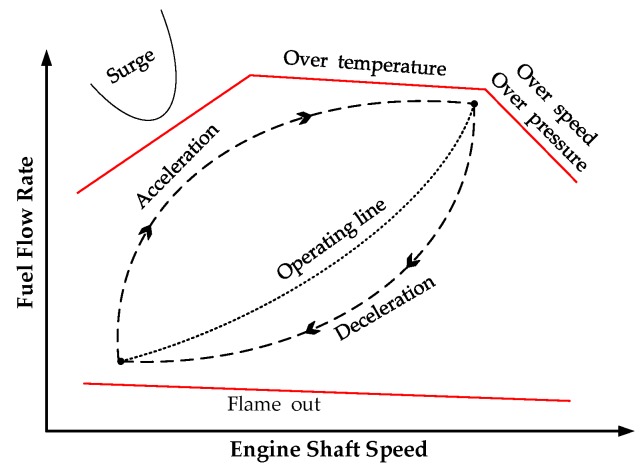
Operation limits for engine control.

**Figure 6 sensors-20-00345-f006:**
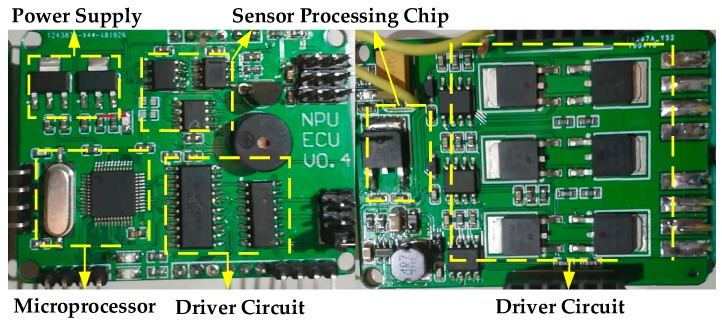
The printed circuit board (PCB) of the designed ECU.

**Figure 7 sensors-20-00345-f007:**
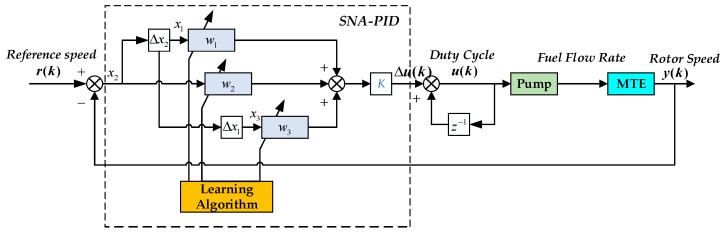
Block diagram of MTE system with single neural adaptive proportional–integral–derivative (SNA-PID) controller.

**Figure 8 sensors-20-00345-f008:**
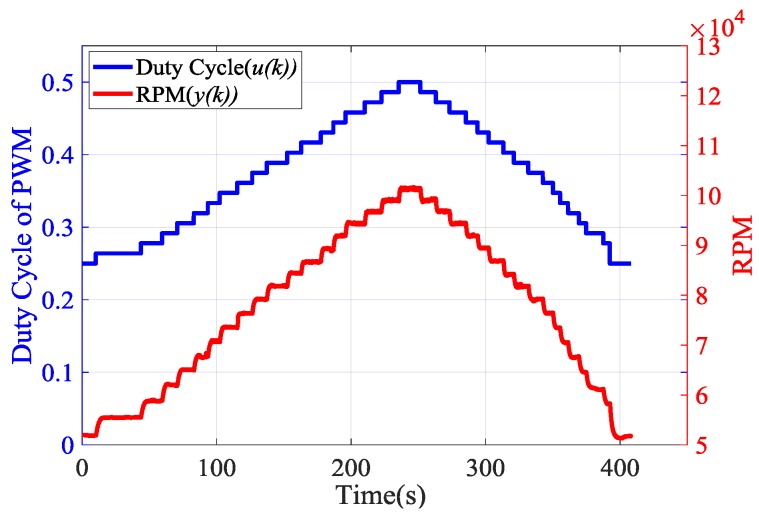
Rotation speed and the duty cycle of pulse width modulation (PWM).

**Figure 9 sensors-20-00345-f009:**
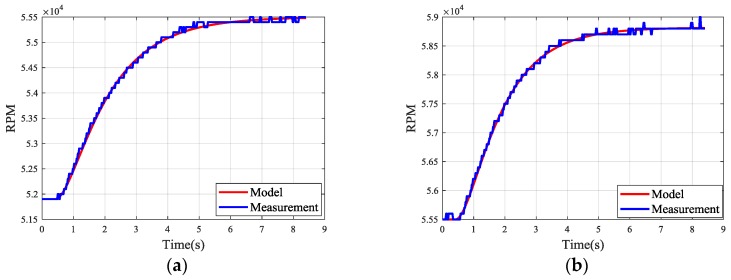
Comparison between real and simulated response. (**a**) 51,900–55,500; (**b**) 55,500–58,900; (**c**) 58,900–61,900; (**d**) 61,900–65,000; (**e**) 65,000–67,800; (**f**) 67,800–70,800.

**Figure 10 sensors-20-00345-f010:**
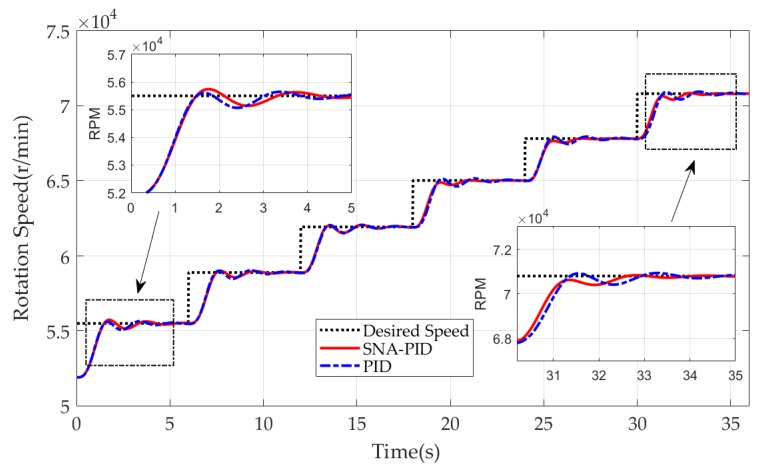
Tracking performance comparison between SNA-PID control and conventional PID control.

**Figure 11 sensors-20-00345-f011:**
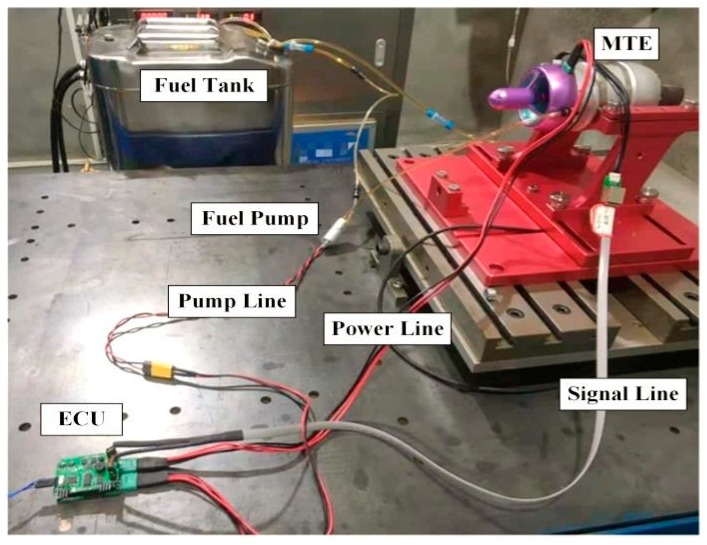
Experimental setup.

**Figure 12 sensors-20-00345-f012:**
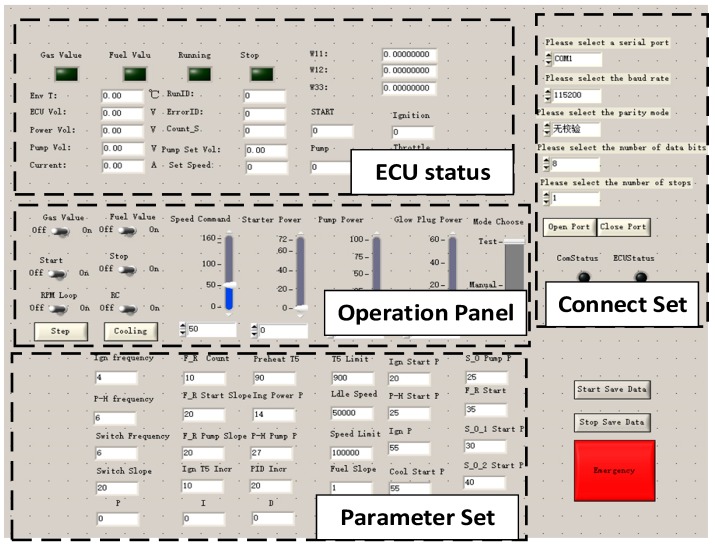
The user interface for experiments.

**Figure 13 sensors-20-00345-f013:**
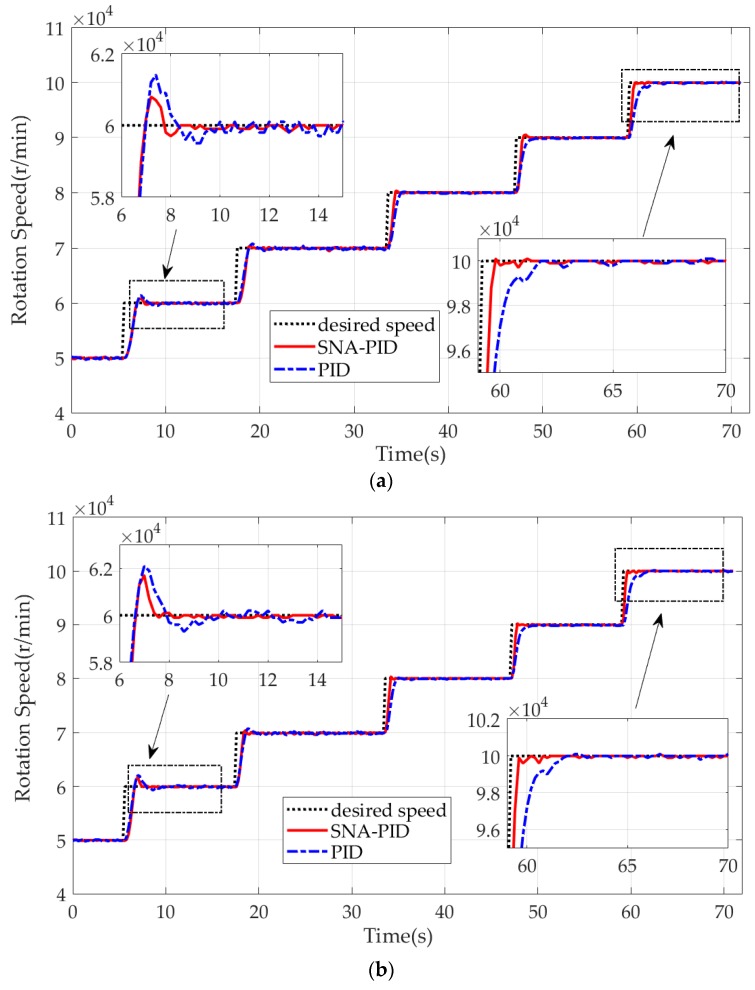
The rotor speed with SNA-PID control vs. conventional PID control. (**a**) Case I; (**b**) Case II.

**Figure 14 sensors-20-00345-f014:**
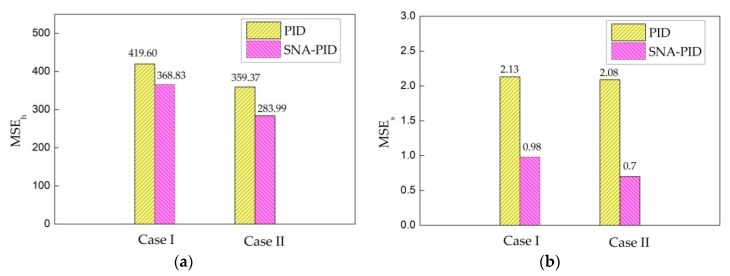
Mean square error (MSE) of the two controllers’ speed tracking. (**a**) The MSE during the whole process; (**b**) the MSE in steady-state.

**Figure 15 sensors-20-00345-f015:**
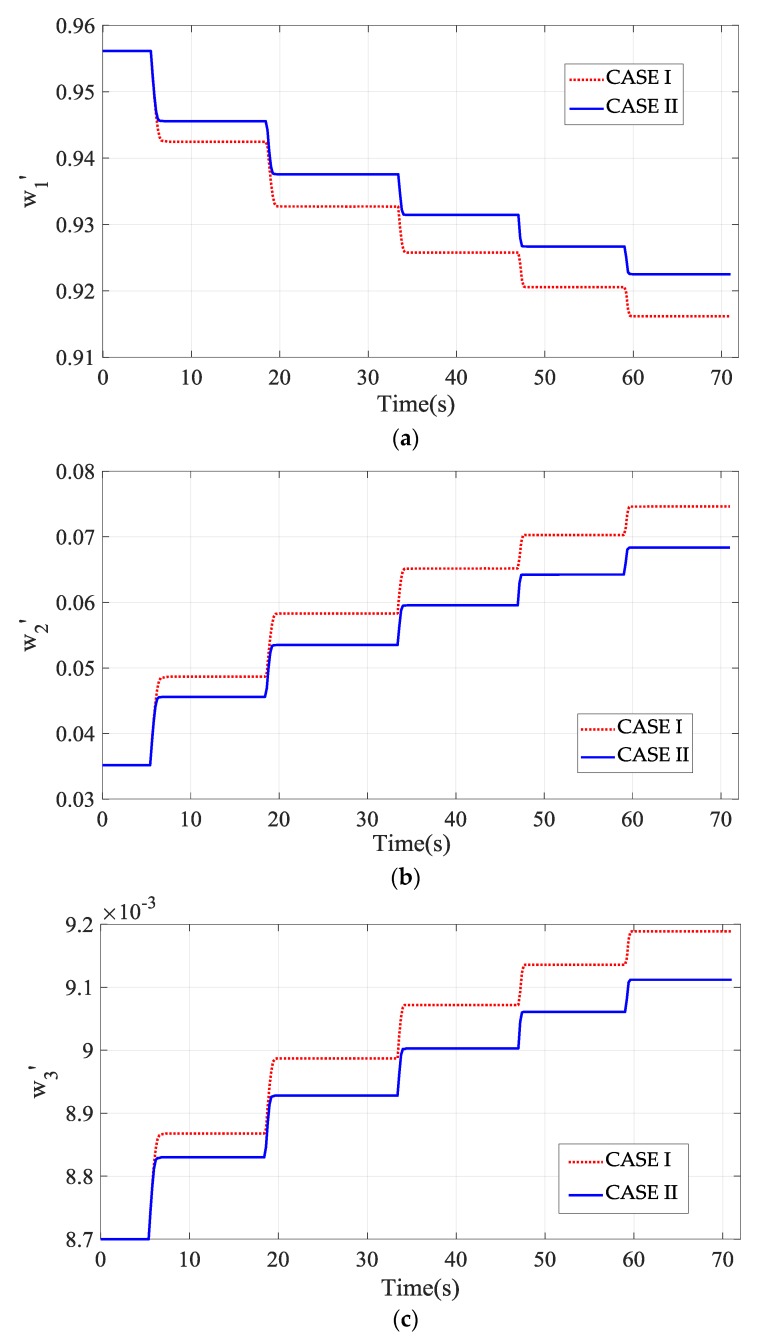
The time history of three weights in the two cases. (**a**) $w_{1}^{\prime }$; (**b**) $w_{2}^{\prime }$; (**c**) $w_{3}^{\prime }$.

**Table 1 sensors-20-00345-t001:** Some basic engine specifications of MTE.

Parameters	Values
Maximum thrust Maximum rotor speed	58.8 N 165,000 RPM
Maximum exhaust temperature	1003 °C
Thrust-to-weight ratio	7.1
Idle speed	50,000 RPM

**Table 2 sensors-20-00345-t002:** The estimated parameters of the engine model at specific speed ranges.

Speed Range (RPM)	$a_{0}$	$a_{1}$	$a_{2}$	$b_{0}$	Fitting Percent (%)
51,900–55,500	1.6165	3.0011	1	58.2171	95.70
55,500–58,900	1.9484	3.1677	1	64.6482	94.05
58,900–61,900	2.1255	2.8148	1	69.0217	95.82
61,900–65,000	2.7681	3.6870	1	89.4422	96.66
65,000–67,800	3.1058	3.8265	1	92.5317	94.30
67,800–70,800	3.0699	3.2701	1	93.0393	89.89

**Table 3 sensors-20-00345-t003:** Performance comparisons between SNA-PID and conventional PID.

Speed Range (RPM)	Rising Time (s)		Setting Time (s)		Overshoot	
	SNA-PID	PID	SNA-PID	PID	SNA-PID	PID
51,900–55,500	1.33	1.31	3.05	2.83	6.67%	2.78%
55,500–58,900	1.32	1.26	2.89	2.90	3.24%	3.24%
58,900–61,900	1.19	1.21	2.72	2.75	3.03%	4.00%
61,900–65,000	1.21	1.23	2.46	2.75	0%	3.87%
65,000–67,800	1.20	1.21	2.46	2.73	0%	4.29%
67,800–70,800	1.16	1.21	2.36	2.75	0%	4.00%

**Table 4 sensors-20-00345-t004:** Different operating conditions for experiments.

Speed Range (RPM)	Case I (Narrow Range)		Case II (Wide Range)	
	$\Delta u_{ac}$	$\Delta u_{dc}$	$\Delta u_{ac}$	$\Delta u_{dc}$
50,000–60,000	10	8	20	15
60,000–70,000	12	8	25	15
70,000–80,000	20	18	40	30
80,000–90,000	40	20	80	40
90,000–100,000	40	22	80	45

**Table 5 sensors-20-00345-t005:** Performance comparisons for Case I.

Speed Range (RPM)	Rising Time (s)		Setting Time (s)		Overshoot	
	SNA-PID	PID	SNA-PID	PID	SNA-PID	PID
50,000–60,000	1.40	1.46	2.20	2.52	8.01%	14.10%
60,000–70,000	1.30	1.33	1.60	2.82	5.01%	8.03%
70,000–80,000	0.90	1.32	0.94	1.43	3.01%	1.00%
80,000–90,000	0.79	1.32	0.84	1.60	5.00%	0%
90,000–100,000	0.63	1.53	0.67	2.40	1.00%	0%

**Table 6 sensors-20-00345-t006:** Performance comparisons for Case II.

Speed Range (RPM)	Rising Time (s)		Setting Time (s)		Overshoot	
	SNA-PID	PID	SNA-PID	PID	SNA-PID	PID
50,000–60,000	1.21	1.18	1.85	3.40	17.00%	21.00%
60,000–70,000	0.92	1.07	1.20	2.87	5.01%	8.03%
70,000–80,000	0.73	1.2	0.80	1.60	3.03%	1.00%
80,000–90,000	0.70	1.35	0.74	1.70	2.01%	0%
90,000–100,000	0.54	1.53	0.57	2.34	0%	1.00%

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
