# Peer review of "Single Neural Adaptive PID Control for Small UAV Micro-Turbojet Engine"

_sensors, 2020, doi:10.3390/s20020345_

Round 1
Reviewer 1 Report
Presented article is up to date. Properly deal with the design of a simple regulator. Simplicity is very important in aviation (not only in the case of reduces the computational load and implementation of the algorithm on low-cost hardware, as the authors mention) and this is confirmed by well-known airline personalities and practices. They reject the complex ideas, methods and solutions of scientific research institutions - universities. In the abstract, I would like to express the percentage accuracy of the qualitative advantage of the proposed regulator (in the article it is about increasing the performance and not the quality of the regulator, what is the difference between them?). At the beginning of the introduction, I would like to provide further relevant and significant references to the literature used, for example: “The fuel flow rate and the nozzle area changes are the most important manipulated input variables” and so on. (expand the list of references). Sort references to references from the beginning of number 1 consecutively. In the introduction, various methods, allegories of control of turbocharger engines, e.g. [10-11]. However, in references [12-19] it is necessary to add in more detail what is the benefit, advantage, disadvantage of the control system. And compared to the control algorithm proposed by the authors, this is what makes them choose this way. References [26-28] describe in more detail the benefits of the industry's environment. If you need to derive the relationships needed to design your new regulator, you need to add the source links. Even in the case of jet engine modeling, it is necessary to add the specific methods and references used in the experimental identification.
You write: “It is difficult to realize the normal NN on low-cost hardware such as a microcontroller unit (MCU)”, I ask the authors to analyze – defined in more detail, in this case unprofessionally sounding terms, normal NN and low cost hardware.
Positive and greatest contribution of this paper is experimental verification of designed regulators in laboratory conditions. The authors openly and directly describe the used components of the control system and the laboratory stand, which I consider a positive contribution. However, the article is then very broad-spectrum and does not focus only on the currently described topic. And other questions arise in this respect, for example:
- why you are currently using the STM32 microcontroller,
- what is the smallest step in managing fuel supply, respectively. What is the smallest value you can change the engine rotor speed? is it sufficient to control the engine speed without measuring the input value of the engine fuel flow input? PWM as input signal acquires what values specifically? as in Fig. 8 - 0.25 to 0.5?
- the fuel pump is how you correctly write the most important actuator, is your own production? also used on other engines?
- how is the engine starting process controlled? what are fuel and gas valves? and when are they used? in the article you write, again unprofessional and inaccurate: "With the speed and temperature information, the ECU generates various control signals for switching magnetic valves or regulating the fuel flow rate". I cannot use various control signals, I have to exactly which signal controls which valve and when. Gas valve doesn't go like a completely different branch? Does the gas engine start? And then how do I manage the fuel supply? page 6 - below figure 5 - start-up - fuel valves? both valves on? gas i fuel? or just fuel?
There are formal shortcomings in the article:
- fig4 edit - after enlarging the picture some lines are crooked and unconnected completely - unprofessional, not suitable for such a quality magazine
- under equation 15, the sentence in MATLAB to estimate the transfer function in equation (16). about me to be a 15th equation
- Chapter 4.2 - weights NN w1 to 3 are a different letter
- table 2 - the designation "s" is typical for Laplace transformation and its operator, so for time I recommend using "sec"
- picture 12 poor quality
- Table 6 does not exist - check references in the text
- figure 12a does not exist - check for bad links
It is well written in the article that in the case of FADEC it is very important to record the parameters, in your case of speed, is there a backup? How is the failure of the engine running?
The disadvantage of the proposed control algorithm is the experimental estimation of the limits (how were the values in Table 3 been determined?). It is unusual for controller simulations to exhibit worse performance than in real laboratory operation. The mathematical models obtained in Chapter 4.1 are not verified. There is no numerical expression of the accuracy of the transmission functions obtained. The article contains only a graphical comparison of modeling results and a brief commentary.
The greatest benefits are presented experiments and their results in the last chapters, which are a great bonus for professionals working with the same complex system. Appropriate and detailed graphically and quantitative indicators based on results indicate the suitability of the proposed algorithm. The non-existent and mixed references to the tables and figures in chapter 5.2 caused doubts, questions, misinterpretation and a complicated overview of the results obtained. Why, at maximum modes, acceleration is almost as desired, i. very fast? Isn't it going to be a heavy load for the engine?
The article is very interesting with scientific and practical contribution. In the future, I recommend to compare it with other management methods mentioned in the introduction. After a thorough correction, analysis and appropriate explanation of the professional and practical questions posed, I recommend accepting and publishing the article.
Reviewer 2 Report
The paper addresses a very important and up-to-date aspect of the automatic control of jet engines, improving the performance of their FADEC by implementing the modern method of neural network based control.
The abstract is well written and offers an appropriate description of the paper. The investigation methods and the model issuing, as well as problems' solving, are well described and explained; results and conclusions clearly and accurate presented. Very good graphics; figures, tables and diagrams are appropriate included in the paper's content.
Suggestion to the authors:please, be more specific regarding the way you have determined the mathematical model of the engine (the transfer function mentioned in section 4.1). Is it the result of your own studies, or was it taken from other sources? Usually, a single-spool jet engine has a first order transfer function and you have described (Eq. 15 pp.10) a second order function. In the same paragraph, 6-th row, the mentioned Eq. (16) probably refers to Eq (15).
Round 2
Reviewer 1 Report
All comments received and incorporated. High quality research and article. Well thank you.